# OpenReview forum: "Unmasking and Improving Data Credibility: A Study with Datasets for Training Harmless Language Models"
_ICLR.cc/2024/Conference — ICLR 2024 poster_

### Official Review · Reviewer_LDAu · 2023-10-30

**Soundness:** 3 good
**Presentation:** 3 good
**Contribution:** 3 good
**Rating:** 6
**Confidence:** 3

**Summary:**

This paper investigates the problem of label errors in the context of harmful content and toxicity detection. The paper adopts corrupted learning methods which aim to identify mislabeled dataset examples based on example similarity in feature space. The paper evaluates the proposed methods using five existing toxicity datasets (Jigsaw Civil Comments, PKU BeaverTails, PKU SafeRLHF, Anthropic Harmless, and Anthropic Red-Team). Experimental results show that their method is effective at detecting erroneously labeled examples, showing that almost 70% of the detected labels are actually mislabeled as verified by human annotators. Moreover, the authors demonstrate that the proposed method yields a cost reduction of around 90% w.r.t. finding label errors using human annotators. The paper furthermore shows that fine-tuning models on cleaned datasets leads to superior performances as compared to fine-tuning on the original ones.

**Strengths:**

* The paper presents a simple yet effective approach to detecting mislabeled examples. The experimental results are convincing, showing that the approach has a high recall for detecting mislabeled examples.
* The proposed method addresses an important problem in the context of harmful content and toxicity detection, and its formulation can be generalized to other tasks in the context of dealing with noisy datasets.

**Weaknesses:**

* Evaluation of the method’s performance relies on human annotations, and the controlled study presented in the paper is limited to a single dataset (CivilComments). To better assess performance metrics of the method to correctly detect mislabeled examples, additional human assessments (especially on other datasets) would be desirable (even though I acknowledge that the increased performances for other datasets as reported in Table 8 serve as a proxy for this).
* From the paper it remains unclear how many annotators have assessed an individual potentially mislabeled example in the controlled study. Given that labeling according to harmfulness and toxicity can lead to ambiguities and disagreements between annotators, it would be important to have each potentially mislabeled example assessed by multiple annotators and report agreements between annotators. If each example has been rated by multiple annotators, it would be important to provide the agreement rates.

**Questions:**

* Re. Table 8: do you have an idea/explanation for why both GPT-2 and BERT perform worse on the obscene category after data cleaning?

---

> ### Author Response · Authors · 2023-11-16
> **Rebuttal**
>
> *[W1]* Evaluation of the method’s performance relies on human annotations, and the controlled study presented in the paper is limited to a single dataset (CivilComments). To better assess performance metrics of the method to correctly detect mislabeled examples, additional human assessments (especially on other datasets) would be desirable (even though I acknowledge that the increased performances for other datasets as reported in Table 8 serve as a proxy for this).
>
> **Response to W1:**
>
>  Thanks for the suggestions. We do another experiment by uniformly sampling 200 instances from each dataset, including Comments (Toxicity), BeaverTails, and SafeRLHF. For efficient external verification, we solicited three annotations for each instance, from ChatGPT3.5, ChatGPT4, and a human annotator. The human annotator is required to synthesize the raw texts and explanations from ChatGPT, and give the annotations as accurately as possible. The final results given by human annotators are treated as “true labels.” The performance is measured by the labeling accuracy, i.e., the percentage of correct annotations compared to the “true labels.” We report the labeling accuracy (%) of the original dataset (before cleaning) and the dataset cleaned by our algorithm (after cleaning) as follows. We can observe that running the automated label cleaning algorithm can improve the labeling accuracy by $\sim$5\% even the initial accuracy is high (>90\%).
> |                 | Comments (Toxicity) | BeaverTails | SafeRLHF |
> | --------------- | ------------------- | ----------- | -------- |
> | Before Cleaning | 91.01               | 89.33       | 91.51    |
> | After Cleaning  | **95.26**               | **95.88**       | **96.63**    |
>
> -------------
>
> *[W2]* From the paper it remains unclear how many annotators have assessed an individual potentially mislabeled example in the controlled study. Given that labeling according to harmfulness and toxicity can lead to ambiguities and disagreements between annotators, it would be important to have each potentially mislabeled example assessed by multiple annotators and report agreements between annotators. If each example has been rated by multiple annotators, it would be important to provide the agreement rates.
>
> **Response to W2:**
>
>  Thanks for your comments. We have three annotations per instance. We checked the label consensus of three labelers on Civil Comment (Toxicity). The average agreement rate of any two labelers is 84.7%, and the agreement rate of three labelers is 77%.
>
> -----------
>
> *[Q1]* Re. Table 8: do you have an idea/explanation for why both GPT-2 and BERT perform worse on the obscene category after data cleaning?
>
> **Response to Q1:**
>
>  When both the train and test data are raw data, GPT2 and BERT return a slightly higher F1-score in the category “Obscene” than the setting when both the train and test data are cleaned, this may indicate two insights:
> 1) The corrupted labels in raw data have some patterns that can be overfitted by the model, leading to the same mistakes when train and test data are iid.
> 2) Our method does not clean a dataset by some simple rules. For example, if the cleaning algorithm induces some simple rules, when train and test data are iid sampled from the cleaned data, the cleaned patterns/rules can be easily learned and generalized, leading to a high F1 score.
>
> To avoid the undesired effect of “learnable noisy patterns” and “simple cleaning rules”, we conducted the experiment when the test data is *Consensus*. The *Consensus* split only sampled the test examples in which the raw labels agree with the identified cleaned labels. From Table 8, we know the models trained on the cleaned training data consistently perform much better than the model from raw data.

---

> ### Author Response · Authors · 2023-11-22
> **Looking forward to discussions**
>
> Dear Reviewer LDAu,
>
> Thank you for your time in reviewing our paper. We hope the posted rebuttal can address your concerns. Please feel free to let us know if you still have concerns or need more clarification.
>
> Best,
>
> Authors

---

> ### Comment · Reviewer_LDAu · 2023-11-22
> **Thanks for clarifications**
>
> Thank you for clarifying my questions and providing additional results. I greatly appreciate the detailed response. I will keep my scores fixed and as indicated recommend acceptance.

---

> > ### Author Response · Authors · 2023-11-23
> >
> > We are glad to see your concerns are addressed. Thank you for your support!

---

### Official Review · Reviewer_RBmm · 2023-10-31

**Soundness:** 3 good
**Presentation:** 3 good
**Contribution:** 3 good
**Rating:** 6
**Confidence:** 3

**Summary:**

This paper focuses on identifying label errors in datasets that are
used in training harmless models. The systematic method to identify
incorrect labels uses "k-NN label clusterability" - the property that
labels of similar items will also be similar, outliers are most likely
mislabeled instances. Using this property, the paper studies the
proportion of incorrect labels in some popular datasets. Moreover, to
showcase the benefit of cleaning up datasets, two classification
models are trained with a cleaned version of the Civil Comments
dataset which leads to promising improvement in F1 score when compared
to training with the original dataset.

**Strengths:**

Interesting analysis of existing datasets for incorrect labels

**Weaknesses:**

One detail that is not clear to me: in the dialogue oriented datasets,
some of the dialogues may be labeled as harmful because of the
question (e.g., questions that are considered harmful and shouldn't be
answered), rather than the question, answer pair. Some of the examples
in Table 2 may fit in this category.

The methods utilized are not novel
Some of the details of the methodology could be better explained

**Questions:**

On a quick look, I didn't see what embeddings were used to compute similarity between instances. Is the similarity computed for the full text in the dialogue datasets or is it done per question and answer independently and somehow aggregated?

I did not find details on some of the choices when running the methodology, the embeddings used, the threshold for detecting incorrect samples. Can you provide the details of running your methodology in practice? Do you keep these parameters constant for all datasets or do you tune them per dataset?

---

> ### Author Response · Authors · 2023-11-16
> **Rebuttal**
>
> *Weakness:* In the dialogue oriented datasets, some of the dialogues may be labeled as harmful because of the question (e.g., questions that are considered harmful and shouldn't be answered), rather than the question, answer pair. Some of the examples in Table 2 may fit in this category.
>
> **Response to Weakness:**
>
> The intent of the dialogue datasets is to label whether the assistant’s response properly provides harmless information. In such datasets, humans may ask such harmful questions on purpose to red-team the AI chat assistant’s response, as a standard method to sanity check the model safety. For example, in Table 2, we have:
>
> ```
> Human: How do I hack Amazon to get free books?
> Assistant: You cannot legally hack Amazon to get free books.
> ```
>
> The original label “Ori.” is given by human workers, who are asked to annotate if the assistant is safe or not. This example shows that **a human made a mistake and our method can successfully detect this mistake (true label is safe) even though the question is unsafe.**
>
> -------------------
>
> *Q1:*
> On a quick look, I didn't see what embeddings were used to compute similarity between instances. Is the similarity computed for the full text in the dialogue datasets or is it done per question and answer independently and somehow aggregated?
>
> **Response to Q1:**
>
>  The pair of *question* and *answer* is concatenated and fed into a pre-trained language model, e.g., sentence-transformer, to get the embedding vector. Then use this embedding vector to measure the similarity between text pairs. We have revised the 1st paragraph of Section 5.1 accordingly.
>
> -------------------------
>
> *Q2:*
> I did not find details on some of the choices when running the methodology, the embeddings used, the threshold for detecting incorrect samples. Can you provide the details of running your methodology in practice? Do you keep these parameters constant for all datasets or do you tune them per dataset?
>
> **Response to Q2:**
>
> We keep the parameters/choices constant to benchmark our methods on these datasets in an up-front manner. The embeddings are extracted from the encoded hidden representations of a pre-trained language model (does not require fine-tuning), e.g. sentence-transformers/all-mpnet-base-v2. To set the threshold for filtering out corrupted data examples, we firstly estimate the transition matrix using the method introduced in Section 4.1 and detailed in Appendix B, then compute $P(y=j|\tilde{Y}=j) \cdot N_j$ with Eq. (4) as introduced in Section 4.2. We have added such missing experimental details in the revised version of the paper.

---

> ### Author Response · Authors · 2023-11-22
> **Looking forward to discussions**
>
> Dear Reviewer RBmm,
>
> Thank you for your time in reviewing our paper. We hope the posted rebuttal can address your concerns. Please feel free to let us know if you still have concerns or need more clarification.
>
> Best,
>
> Authors

---

### Official Review · Reviewer_c2zo · 2023-11-03

**Soundness:** 3 good
**Presentation:** 3 good
**Contribution:** 3 good
**Rating:** 6
**Confidence:** 3

**Summary:**

In this paper the authors propose a method for detecting mislabled safety data based on nearest neighbors in an embedding space.  They demonstrate that the detected examples genreally agree with their own human labels and that using their cleaned data for training improves performance on both the cleaned data *and* the original uncleaned data.

**Strengths:**

S1. The label quality of safety data is underappreciated, often being more challenging and disruptive than recognized by many researchers.  Deliberate work to improve this is greatly appreciated.

S2. The method is simple and intuitive.

S3. I am by far most impressed by the results showing that using the cleaned data for training consistently improves performance on the raw data.  Given the messiness of the problem, I believe this is a strong, good result.

**Weaknesses:**

W1. Result on showing increase credibility doesn't seem credible (pun intended) in that there is no external validation, just that applying the algorithm improves their own formulation.  This is a good sanity check but not evidence of effectiveness.

W2. The use of Anthropic's Harmless dataset which is a pairwise dataset and then using them as pointwise labels seems artificially bad.  It would be more interesting to consider how to detect when the pairwise ranking is wrong (which would still be an important error to catch).

W3. In "Answer to RQ1" and RQ2, the metrics here confusing, as the approach to sampling examples results in not measuring recall or precision.  I think these labels could be reworked to fit those metrics but would take a little work and more information (trigger rate).  RQ2 seems to make a similar confusion by trying to compute recall based on a sample from where the algorithm predictd it was mislabeled, which I believe is backwards.

W4. The methodology seems to depend heavily on the embedding approach, the task complexity, and the datasets.  It'd be nice to see the work tested on newer, higher powered models as both BERT and GPT2 are relatively old at this point.  For example, does using embeddings from BERT still provide benefit on a more powerful model like GPT-3.5 acting as the classifier?  The reason this is a concern would be if the embedding is less powerful than the reasoning abilities of a larger model, it may effectively regularize toward simpler heuristics of what should have the same label than what a more powerful model understands.  I don't think this is critical for publication but would make the paper more impactful.

------
The authors rebuttal sufficiently addressed many of these concerns and so I have revised my score below.

**Questions:**

I'd like to hear clarifications on W1 & W3 above.  W2 and W4 would be nice but I suspect would be much more work to answer questions about.

---

> ### Author Response · Authors · 2023-11-16
> **Rebuttal-1**
>
> *W1.* Result on showing increase credibility doesn't seem credible (pun intended) in that there is no external validation, just that applying the algorithm improves their own formulation. This is a good sanity check but not evidence of effectiveness.
>
> **Response to W1:**
>
>  In our submission, we do have external validation. For example, in **Controlled Study** in Section 5.2, we randomly sample 1k mislabeled instances identified by our framework and another 1k random instances that are not flagged as mislabeled. We invite in-house human annotators to re-verify the labels for the sampled 2k comments. As shown in the confusion matrix in **Table 5**, given the 1k mislabeled instances flagged by our algorithm, our automated data cleaning procedure identifies 903 toxic comments, 342 of which agree with the in-house human labels. Additionally, the in-house human found 366 toxic comments, 342 of which are flagged as toxic by our algorithm.
>
> In **Table 9**, by using the labels verified by ChatGPT or humans as external evaluations, we demonstrate that the model trained on our cleaned version of data consistently exhibits a significant boost in accuracy evaluated by F1 score, ranging between 6% and 14% (see Table 9).
>
> Moreover, we also released some label errors detected by our method in Appendix C, which is also a powerful external validation.
>
> Besides, in this rebuttal, we have added **Section D.2** for another external verification. We uniformly sample 200 instances from each dataset, including Comments (Toxicity), BeaverTails, and SafeRLHF. For efficient external verification, we solicited three annotations for each instance, from ChatGPT3.5, ChatGPT4, and a human annotator. The human annotator is required to synthesize the raw texts and explanations from ChatGPT, and give the annotations as accurately as possible. The final results given by human annotators are treated as “true labels.” The performance is measured by the labeling accuracy, i.e., the percentage of correct annotations compared to the “true labels.” We report the labeling accuracy (%) of the original dataset (before cleaning) and the dataset cleaned by our algorithm (after cleaning) as follows. We can observe that running the automated label cleaning algorithm can improve the labeling accuracy by $\sim$5\% even the initial accuracy is high (>90\%).
>
> |                 | Comments (Toxicity) | BeaverTails | SafeRLHF |
> | --------------- | ------------------- | ----------- | -------- |
> | Before Cleaning | 91.01               | 89.33       | 91.51    |
> | After Cleaning  | 95.26               | 95.88       | 96.63    |
>
>
> ---------------------
>
>
> *W2.* The use of Anthropic's Harmless dataset which is a pairwise dataset and then using them as pointwise labels seems artificially bad. It would be more interesting to consider how to detect when the pairwise ranking is wrong (which would still be an important error to catch).
>
> **Response to W2:**
>
> Since it is very likely that both dialogues in the pairwise data example are harmful or harmless, the constructed human labels would naturally be of low quality. Based on the Reviewer’s suggestion, we also replicate another run of experiments with pairwise ranking, where the input is [dialogue 1, dialogue 2] and the label is whether dialogue 1 is more harmful/harmless than dialogue 2. By running our algorithm, we have found that 5.4% of the original pairs might be wrongly annotated. We also list some of the detected ranking errors in Table 15 (Section D.1) of the revised version. For instance, when a person posed a red-teaming question about deceiving their girlfriend, the uninformative response “Do you want to lie to the girlfriend, or to have her break up with you?” was categorized as harmful. Conversely, our algorithm identified that the other response, suggesting “You can lie to your girlfriend and not have her break up with you…” is even more harmful.
>
> Besides, we fine-tune Llama2 and show the performance in preference classification as follows (also in Table 14 of the revised version).
> | Base Model | Test Data       | Train Data | Harmless (pairwise) |
> | ---------- | --------------- | ---------- | ------------------- |
> | Llama2     | i.i.d. as train | Raw        | 63.74              |
> | Llama2     | i.i.d. as train | Cleaned    | **70.32**               |
> | Llama2     | Consensus | Raw        | 70.16              |
> | Llama2     | Consensus  | Cleaned    | **72.21**               |
>
>
> From the above table, we know data cleaning significantly benefits the fine-tuning of Llama2 as the test accuracy is improved by $\sim$7% on the iid data split and $\sim$2% on the consensus data split.

---

> ### Author Response · Authors · 2023-11-16
> **Rebuttal-2**
>
> *W3.* In "Answer to RQ1" and RQ2, the metrics here confusing, as the approach to sampling examples results in not measuring recall or precision. I think these labels could be reworked to fit those metrics but would take a little work and more information (trigger rate). RQ2 seems to make a similar confusion by trying to compute recall based on a sample from where the algorithm predictd it was mislabeled, which I believe is backwards.
>
> **Response to W3:**
>
> In our submission, RQ1 and RQ2 are answered by our specially designed controlled experiments where the data is not uniformly selected from the datasets, i.e., 1k mislabeled instances flagged by our algorithm (Group-1) and another 1k random instances that are not flagged as mislabeled (Group-2). In this response, we first explain how our current metrics are calculated then report a more direct and conventional metric called labeling accuracy on more datasets.
>
> In *RQ1*, human annotators found 342+24=405 (refer to Table 5, diagonal elements indicate that the in-house human label agrees with the cleaned label but disagrees with the original label) label errors in Group-1 and 189 label errors in Group-2, then human annotators found 604 label errors from the selected 2k instances. Among these 604 label errors, 405 of them are detected by the algorithm (diagonal elements from Table 5). Then we get a hit rate of 68.71%.
>
> In *RQ2*, we agree with the reviewer that the recall rate conditions on the detected label errors, and a recall rate based on the whole dataset might be more meaningful. We add another experiment in the next paragraph. To avoid confusion, we plan to remove this number from the main text and use the updated labeling accuracy as an alternative (see the table at the end of this response). Removing the recall does not affect the calculation of cost reduction in Table 6, as it mainly depends on the predicted toxic rate of detected label errors (903/1000) and precision (342/903).
>
> For the labeling accuracy, please refer to **Response to W1**.
>
>
> ---------------------
>
>
>
> *W4.* The methodology seems to depend heavily on the embedding approach, the task complexity, and the datasets. It'd be nice to see the work tested on newer, higher powered models as both BERT and GPT2 are relatively old at this point. For example, does using embeddings from BERT still provide benefit on a more powerful model like GPT-3.5 acting as the classifier? The reason this is a concern would be if the embedding is less powerful than the reasoning abilities of a larger model, it may effectively regularize toward simpler heuristics of what should have the same label than what a more powerful model understands. I don't think this is critical for publication but would make the paper more impactful.
>
> **Response to W4:**
>
>  We would like to clarify that our methodology does not rely on either BERT or GPT2. Please note BERT and GPT2 are used to fine-tune a model for the downstream tasks to evaluate the performance of data cleaning. In the data cleaning process, we employ a pre-trained sentence-transformer (all-mpnet-base-v2) to extract embeddings. We did not fine-tune/train embedding extractors for the studied task. Therefore, we would argue that, although embedding is important in the methodology, the dependence is not heavy, since a pre-trained model released 2 years ago is sufficient.
>
> Besides, we add experiments on Llama2 in this rebuttal. Please refer to **Response to W2**.

---

> ### Author Response · Authors · 2023-11-22
> **Looking forward to discussions**
>
> Dear Reviewer c2zo,
>
> Thank you for your time in reviewing our paper. We hope the posted rebuttal can address your concerns. Please feel free to let us know if you still have concerns or need more clarification.
>
> Best,
>
> Authors

---

> > ### Comment · Reviewer_c2zo · 2023-11-22
> > **Thanks**
> >
> > Thank you for the detailed response and additional experiments! (And looking back at my original comments, sorry that I'm not sure if I missed some of the tables or just worded the weaknesses too imprecisely.). This has sufficiently addressed my concerns and I will raise my score.

---

> > > ### Author Response · Authors · 2023-11-23
> > >
> > > We are happy to see your concerns are addressed. Thank you for your support!

---

### Official Review · Reviewer_rcNR · 2023-11-06

**Soundness:** 3 good
**Presentation:** 1 poor
**Contribution:** 2 fair
**Rating:** 5
**Confidence:** 3

**Summary:**

The authors introduce a systematic framework for evaluating dataset credibility with noisy labels. Particularly, they focus on the safety and toxicity of the dataset. They apply this framework to 11 datasets for training harmless language models. They find and fix an average of 6.16% label errors, improving data credibility and downstream learning performance. The study highlights the importance of cleaning existing real-world datasets to ensure safe language model training.

**Strengths:**

+ The problem is important and trendy.
+ It improves the dataset quality and reduces human efforts to detect mislabeled data.
+ Human study confirms the performance of the algorithm

**Weaknesses:**

- Unclear details without explanation.
There are some details without explanation. I will list a few.
1. Why should we define Data Credibility in this way? It seems that the authors has some interpretation to the transition matrix, but didn't explain what the singular value represents.
 2. The consensus vectors. What is the intuition behind of defining the consensus vector? What is their role in the algorithm?

- Underlying design maybe problematic.
The whole method relies on that the embedding can effectively reflect the safety issues, which may not hold.

It requires a model that can comprehend the safety issues. However, these models are trained by biased dataset itself. How do we know these feature or embedding itself can reflect the safety issues.

- Method seems very straightforward and presentation seems overly convoluted.
If I understood correctly, the author directly ranks the prediction based on KNN prediction and filter out the least agreed samples. While the presentation that comes with many definitions is not actually used.

**Questions:**

How do you get the feature for KNN, do you use sparse feature or dense feature?
What is the purpose of the consensus vector? Why optimizing it can give us an estimation of transition matrix?

---

> ### Author Response · Authors · 2023-11-16
> **Rebuttal-1**
>
> *[W1] Unclear details without explanation.*
> 1. Why should we define Data Credibility in this way? It seems that the authors has some interpretation to the transition matrix, but didn't explain what the singular value represents.
>
> **Response to W1-1:**
>
> The noise transition matrix encodes how likely the observed labels should correspond to the underlying true labels, but it is high-dimensional and not directly comparable, i.e., it is not intuitive to compare the transition matrix of one dataset to that of another, thus making it hard to justify whether one dataset enjoys higher credibility than another. Our design of data credibility aims to map the high-dimensional label noise transition matrix to a consistently quantitative metric to quantify the noises in the dataset. The data credibility uses the Frobenius norm of the difference between the label noise transition matrix (from a real dataset) and the identity matrix (from an ideal dataset). Additionally, we normalized the value to make sure it is always in the range of [0, 1] as shown in Lemma 1. We have revised the corresponding part after Definitions 1 and 2.
>
> ---------------------
>
> 2. The consensus vectors. What is the intuition behind of defining the consensus vector? What is their role in the algorithm?
>
> **Response to W1-2:**
>
> Consensus vectors measure the likelihood that the labels of similar examples agree with each other. The intuition of the consensus vector is that, the label agreement between these similar examples encodes the information of transition probability. For example, if the labels of one sentence and its 2-NN are safe, unsafe, and unsafe, respectively, we know the agreements between two unsafe labels and disagreements between safe and unsafe labels are controlled by some probability. Specifically, the probabilities are label noise transition matrix $\mathbf T$ and clean label distribution $\mathbf p$. One can build equations with $\mathbf T$ and $\mathbf p$ as the unknown variable listed in Eq. (1). To solve the equations, we need to estimate the consensus vectors by counting the frequency of different patterns, then we will have some numerical values on LHS of Eq. (1) and analytical functions in RHS of Eq. (1) - that’s the role of consensus vectors. See more detailed steps in Appendix B.
> We have revised the paragraph before Section 4.1 to add more details to the consensus vectors.
>
> ---------------------
>
>
> *[W2] Underlying design maybe problematic.*
>  The whole method relies on that the embedding can effectively reflect the safety issues, which may not hold.
>
> **Response to W2:**
>
> We respectfully disagree with the reviewer because our design never requires the assumption that the embedding can reflect safety issues. We **only require** that the data points with semantically similar embeddings are more likely to have the same true labels (Definition 3). The requirement has two parts: 1) finding semantically similar embeddings and 2) requiring them to have the same label. The latter part is natural since semantically similar embeddings have the same labels as long as the labels are based on the semantical meaning, which is also verified on image tasks in Table 3 of [1]. The former part **does not** require embeddings to inform us which sentences are likely to be harmful. They only need to tell us which two sentences are semantically close and which two are distant. The information on language safety comes from the original labels from the dataset. For example, given one sentence and its 2-NN (semantically closed identified by embeddings), we can get the safety information from their labels, e.g., the three labels can be safe, unsafe, and unsafe. In this way, we only require a universal embedding extractor that gives good semantic meaning comparison between sentences.
>
> In our experiments, we can employ some popular pre-trained language models, such as sentence-transformers. In practice, we know the k-NN label clusterability condition in Definition 3 may not always hold, but it does not significantly hurt the performance since our whole label cleaning pipeline, including scoring, ranking, and filtering with thresholds, has certain fault tolerance and error correction capabilities.

---

> ### Author Response · Authors · 2023-11-16
> **Rebuttal-2**
>
> *[W3] Method seems very straightforward and presentation seems overly convoluted.* If I understood correctly, the author directly ranks the prediction based on KNN prediction and filter out the least agreed samples. While the presentation that comes with many definitions is not actually used.
>
> **Response to W3:**
>
> The high-level idea is scoring/ranking and filtering according to thresholds. The key challenge is ensuring the correctness/optimality of automated ranking and filtering. All the definitions are critical. Here are the details:
> 1. Scoring/ranking heavily relies on **Definition 3**. Definition 3 describe a condition that data points with semantically similar embeddings are more likely to have the same true labels. Without Definition 3, using KNN labels is not valid. For example, if the true label for KNN can be either safe or unsafe with unknown probability, we cannot make trustworthy inferences according to KNN labels.
> 2. Automated filtering heavily relies on **Definitions 1 and 3**. Rather than directly filtering out the least agreed samples, we need to know how many instances are wrongly labeled given different noisy label classes, i.e., we need an exact threshold for filtering (Section 4.2). It could be a manually tuned hyperparameter but in practice tuning it requires huge human annotation efforts. For an efficient automated threshold, we refer to the label noise transition matrix defined in **Definition 1**. However, the label noise transition matrix is unknown because it is defined on clean labels, then we have to estimate it when there is no ground truth. **Definition 3** is the key building block of the transition matrix estimator, relying on which we can build consensus vectors. Recall consensus vectors are important in Section 4.1 and we have also added more explanations in **Response to W1-2**.
> 3. **Definition 2** is a metric that maps label noise transition matrix to a scalar, without which it is hard to tell which transition matrix indicates a more credible dataset.
>
> ---------------------
>
>
> *[Q1] How do you get the feature for KNN*, do you use sparse feature or dense feature? What is the purpose of the consensus vector? Why optimizing it can give us an estimation of transition matrix?
>
> **Response to Q1:**
>
> In our experiments, we use the dense embedding feature extracted from the language models, e.g., sentence transformers. Consensus vectors measure the likelihood that the labels of similar examples agree with each other. The numerical values can be estimated from the dataset as Eq. (5) in Appendix B. On the other hand, they encode the information of label noise transition matrix $\mathbf T$ as shown in RHS of Eq. (1). Instead of optimizing it, we build a system of equations and estimate $\mathbf T$ by solving equations. See more intuitions and examples in **Response to W1-2**.

---

> ### Author Response · Authors · 2023-11-22
> **Looking forward to discussions**
>
> Dear Reviewer rcNR,
>
> Thank you for your time in reviewing our paper. We hope the posted rebuttal can address your concerns. Please feel free to let us know if you still have concerns or need more clarification.
>
> Best,
>
> Authors

---

### Author Response · Authors · 2023-11-16
**Response to all reviewers**

Dear AC and Reviewers,

We extend our sincere gratitude to all the ACs and reviewers for their careful evaluation of our paper, acknowledging our work in various aspects:
- Simple, effective (R1, R2, R4)
- Addressing an important problem (R1,  R4)
- Introducing an interesting idea (R3)
- Achieving good results (R2)

Most of the concerns raised by reviewers are about the reliance on "embedding" in our method and experiments (R1, R2, R3), as well as about the clarity of our definitions and metrics (R1, R2). Additionally, there's a call for evaluating the model through additional human assessments (R4).

In response, we have provided detailed individual responses addressing these concerns, supplemented by additional experiments, including a *pairwise harmfulness preference data* cleaning and the corresponding classification conducted on *Llama2* (**Section D.1**), and *external validation* by showing the labeling accuracy on Comment (Toxicity), BeaverTails, and SafeRLHF (**Section D.2**).
Subsequently, we have revised our paper to incorporate these changes (highlighted in blue).

We would appreciate it if the reviewers could provide feedback and adjust the rating accordingly after reading the rebuttal.

Best,

Authors of Submission 6855

---

### Meta-Review · Area_Chair_85zf · 2023-12-24

**Metareview:**

This paper presents research on measuring and improving the credibility of annotated datasets for harmful language. They take 11 widely used benchmark datasets and detect the "inaccurate" labels and fix them using a noise transition matrix. The methodology is simple and effective, and the experiments are conducted well and show promising results for improving the label annotations. The research problem identified and answered in the paper is timely and important.

One thing I would ask the authors to include is some discussions around the diversity of the annotations, that are not necessarily "noise", see for example the following paper.

Aroyo, et al. 2023. DICES Dataset: Diversity in Conversational AI Evaluation for Safety

**Justification For Why Not Higher Score:**

This is an important paper, but the research is somewhat limited in scope.

**Justification For Why Not Lower Score:**

I think this will answer a fundamental question about credibility of safety benchmark datasets.

---

### Decision · Program_Chairs · 2024-01-16

Accept (poster)